# Coalescing hardcore-boson condensate states with nonzero momentum

**C. H. Zhang$^{1\star}$, Z. Song$^{1\dagger}$**

**1** School of Physics, Nankai University, Tianjin 300071, China

$\star$ zhangchuhang@mail.nankai.edu.cn $\dagger$ songtc@nankai.edu.cn

## Abstract

Exceptional points (EPs), as an exclusive feature of a non-Hermitian system, support coalescing states to be alternative stable state beyond the ground state. In this work, we explore the influence of non-Hermitian impurities on the dynamic formation of condensate states in one-, two-, and three-dimensional extended Bose-Hubbard systems with strong on-site interaction. Based on the solution for the hardcore limit, we show exactly that condensate modes with off-diagonal long-range order (ODLRO) can exist when certain system parameters satisfy specific matching conditions. Under open boundary conditions, the condensate states become coalescing states when the non-Hermitian $\mathcal{PT}$-symmetric boundary gives rise to the EPs. The fundamental mechanism behind this phenomenon is uncovered through analyzing the scattering dynamics of many-particle wavepackets at the non-Hermitian boundaries. The EP dynamics facilitate the dynamic generation of condensate states with non-zero momentum. To further substantiate the theoretical findings, numerical simulations are conducted. This study not only unveils the potential condensation of interacting bosons but also offers an approach for the engineering of condensate states.

## 1  Introduction

Recent developments in cold atom experiments provide a versatile platform for realizing various phases of interacting and non-interacting bosonic systems [1–4] . Available experimental setups nowadays allow for the control of both geometry and interactions, so as to investigate the real-time evolution of quantum many-body systems directly with engineered model Hamiltonians [1, 5, 6]. It thus boosts the theoretical predictions of exotic quantum phases in interacting systems, which then might be realized and tested in experiments. Exact solutions for quantum many-body systems are rare, but important for providing valuable insights for the characterization of new forms of quantum matter and dynamic behaviors.

Bose-Einstein condensation (BEC) is one of the most striking manifestations of the quantum nature of matter on the macroscopic scale [7]. It represents a formation of a collective quantum state of free bosons. Intuitively, on-site repulsive interactions should block the formation of BEC under the moderate particle density. A lot of effort has been devoted to investigate and understand the role of particle-particle interactions on the occurrence of BEC [8, 9].

On the other hand, recent years have seen a growing interest in non-Hermitian descriptions of condensed-matter systems [10–26] It has been shown that the interplay between non-Hermiticity and interaction can give rise to exotic quantum many-body effect, ranging from non-Hermitian extensions of Kondo effect [15, 27], many-body localization [21], Fermi surface in coordinate space [28], to fermionic superfluidity [18, 29]. The cooperation between the non-Hermiticity and interaction may lead to rich quantum phases due to the peculiarity of non-Hermitian system.

Exceptional points (EPs), as an exclusive feature of a non-Hermitian system, are degeneracies of non-Hermitian operators [30–33]. The corresponding eigenstates coalesce into one state, resulting in the incompleteness of Hilbert space. The peculiar features around EP have sparked tremendous attention to the classical and quantum photonic systems [34–40].Notably, a coalescing state has an exclusive feature. On the one hand, it is an eigenstate of the Hamiltonian, on the other hand, it has the advantage that it is also a target state for a long-time evolution of various initial states. In this sense, a coalescing state is an alternative stable state beyond the ground state. Given the above rapidly growing fields in experimental and theoretical perspectives, we are motivated to investigate the impact of non-Hermitian impurities on the dynamic formation of condensate states of interacting bosons.

In this paper, we study one-, two-, and three-dimensional extended Bose-Hubbard systems with strong on-site interaction. The exact solution for the hardcore limit shows that there exists condensate modes when the system parameters meet the matching conditions. It also allows us to calculate the correlation function for any size system, so as to prove that condensate states indeed possesses off-diagonal long-range order (ODLRO) [41]. We focus on the impact of non-Hermitian impurities on the dynamic formation of condensate states. For open boundary condition, the condensate states become coalescing states when the non-Hermitian $\mathcal{PT}$-symmetric boundary induces the EPs. The underlying mechanism is revealed by the reflectionless absorption of many-particle wavepacket with resonant momentum by the non-Hermitian boundary. In parallel, the EP dynamics allows the dynamic generation of condensate states with nonzero momentum. We perform numerical simulations for finite-size system to demonstrate and verify the theoretical results. The finding not only reveals the

possible condensation of interaction bosons, but also provides a method for condensate state engineering in an alternative way. The implications of this work are significant for both theoretical and practical applications in the realm of quantum many-body systems and could pave the way for innovative strategies in quantum state manipulation and control.

This paper is organized as follows. In Sec. 2, we introduce the model Hamiltonian and its condensate eigenstate. In Sec. 3, we derive the exact condition for coalescing condensate state. In Sec. 4, we perform the numerical simulations to demonstrate reflectionless scattering of many-particle Gaussian wavepacket at non-Hermitian boundary. In Sec. 5, we present the possibility of dynamically generating condensate state with an arbitrary initial state. Finally, we summarize our results in Sec. 6.

## 2   Model and condensate states

We start our study from a general form of the Hamiltonian on a three-dimensional lattice $N_1 \times N_2 \times N_3$

$$
\begin{aligned}
H = {} & \sum_{\alpha=1}^{3} J_\alpha \sum_{\mathrm{r}} \frac{1}{2} \hat{a}_{\mathrm{r}}^\dagger \hat{a}_{\mathrm{r}+\mathrm{e}_\alpha} + \mathrm{H.c.} + \sum_{\alpha=1}^{3} V_\alpha \sum_{\mathrm{r}} \hat{n}_{\mathrm{r}} \hat{n}_{\mathrm{r}+\mathrm{e}_\alpha} \\
& + \sum_{\alpha=1}^{3} \sum_{\mathrm{r}} \left( \mu_\alpha \hat{n}_{\mathrm{r}} \delta_{1,m_\alpha} + \mu_\alpha^* \hat{n}_{\mathrm{r}} \delta_{N_\alpha,m_\alpha} \right),
\end{aligned}
\tag{1}
$$

where $\hat{a}_{\mathrm{r}}^\dagger$ is the hardcore boson creation operator at the position $\mathrm{r} = m_1 \mathrm{e}_1 + m_2 \mathrm{e}_2 + m_3 \mathrm{e}_3$ ($m_\alpha = 1, 2, ..., N_\alpha$, $\alpha = 1, 2, 3$), satisfying

$$
\left\{ \hat{a}_l, \hat{a}_l^\dagger \right\} = 1, \{ \hat{a}_l, \hat{a}_l \} = 0,
\tag{2}
$$

and

$$
\left[ \hat{a}_j, \hat{a}_l^\dagger \right] = 0, \left[ \hat{a}_j, \hat{a}_l \right] = 0,
\tag{3}
$$

for $j \neq l$, and $\hat{n}_{\mathrm{r}} = \hat{a}_{\mathrm{r}}^\dagger \hat{a}_{\mathrm{r}}$, $\mathrm{e}_\alpha$ is the unit vector for $N_\alpha$. Under the open boundary condition, we define $\hat{a}_{\mathrm{r}+N_\alpha \mathrm{e}_\alpha} = 0$, while $\hat{a}_{\mathrm{r}+N_\alpha \mathrm{e}_\alpha} = \hat{a}_{\mathrm{r}}$ for the periodic boundary condition ($\alpha = 1, 2, 3$).

The parameters are taken as

$$
\begin{cases}
V_\alpha = J_\alpha \cos q_\alpha \\
\mu_\alpha = J_\alpha \frac{e^{iq_\alpha}}{2}
\end{cases},
\tag{4}
$$

with arbitrary real number $q_\alpha$ for the case with open boundary condition, but with $\mathrm{q} = (q_1, q_2, q_3)$, $q_\alpha = 2\pi m_\alpha / N_\alpha$ ($m_\alpha = 1, 2, ..., N_\alpha$, $\alpha = 1, 2, 3$) and $\mu_\alpha = 0$ with periodic boundary condition. When taking $\{N_\alpha\} = (N_1, N_2, N_3) = (N_1, N_2, 1)$ or $(N_1, 1, 1)$, the system reduces to two- or one-dimensional systems.

In the following, we will show that state

$$
|\psi_n\rangle = \frac{1}{\Omega_n} \left( \sum_{\mathrm{r}} \hat{a}_{\mathrm{r}}^\dagger e^{-i\mathrm{q}\cdot\mathrm{r}} \right)^n |0\rangle,
\tag{5}
$$

$$
\Omega_n = \frac{1}{(n!)\sqrt{C_N^n}},
\tag{6}
$$

is an eigenstate of the system, where the vacuum state $|0\rangle = \prod_{\mathrm{r}} |0\rangle_{\mathrm{r}}$, with $\hat{a}_{\mathrm{r}} |0\rangle_{\mathrm{r}} = 0$. In both two cases (also including mixed boundary conditions), the Hamiltonian can be written as the form

$$
H = \sum_{\alpha=1}^{3} \sum_{\mathrm{r}} h_{\mathrm{r}}^\alpha + \sum_{\alpha=1}^{3} V_\alpha \hat{n},
\tag{7}
$$

where the dimer term is non-Hermitian, i.e.,

$$
\begin{aligned}
h_{\mathrm{r}}^{\alpha} = J_{\alpha}[ &\frac{1}{2}\hat{a}_{\mathrm{r}}^{\dagger}\hat{a}_{\mathrm{r}+\mathrm{e}_{\alpha}} + \mathrm{H.c.} \\
&+ \cos(\mathrm{q}\cdot\mathrm{e}_{\alpha})(\hat{n}_{\mathrm{r}}\hat{n}_{\mathrm{r}+\mathrm{e}_{\alpha}} - \hat{n}_{\mathrm{r}} - \hat{n}_{\mathrm{r}+\mathrm{e}_{\alpha}}) \\
&+ \frac{1}{2}\left(e^{i\mathrm{q}\cdot\mathrm{e}_{\alpha}}\hat{n}_{\mathrm{r}} + e^{-i\mathrm{q}\cdot\mathrm{e}_{\alpha}}\hat{n}_{\mathrm{r}+\mathrm{e}_{\alpha}}\right)],
\end{aligned}
\tag{8}
$$

and $\hat{n} = \sum_{\mathrm{r}}\hat{n}_{\mathrm{r}}$ is the total number operator. It is easy to check that

$$
\begin{aligned}
h_{\mathrm{r}}^{\alpha}[e^{-i\mathrm{q}\cdot\mathrm{r}}\hat{a}_{\mathrm{r}}^{\dagger} + e^{-i\mathrm{q}\cdot(\mathrm{r}+\mathrm{e}_{\alpha})}\hat{a}_{\mathrm{r}+\mathrm{e}_{\alpha}}^{\dagger}]\,|0\rangle_{\mathrm{r}}\,|0\rangle_{\mathrm{r}+\mathrm{e}_{\alpha}} &= 0, \\
h_{\mathrm{r}}^{\alpha}\hat{a}_{\mathrm{r}}^{\dagger}\hat{a}_{\mathrm{r}+\mathrm{e}_{\alpha}}^{\dagger}\,|0\rangle_{\mathrm{r}}\,|0\rangle_{\mathrm{r}+\mathrm{e}_{\alpha}} &= 0, \\
h_{\mathrm{r}}^{\alpha}\,|0\rangle_{\mathrm{r}}\,|0\rangle_{\mathrm{r}+\mathrm{e}_{\alpha}} &= 0,
\end{aligned}
\tag{9}
$$

which ensures that

$$
H\,|\psi_{n}\rangle = n\sum_{\alpha=1}^{3}V_{\alpha}\,|\psi_{n}\rangle.
\tag{10}
$$

In addition, state $|\psi_{n}\rangle$ possesses ODLRO due to the fact that the correlation function

$$
\langle\psi_{n}|\,\hat{a}_{\mathrm{r}}^{\dagger}\hat{a}_{\mathrm{r}+\mathrm{R}}\,|\psi_{n}\rangle = e^{-i\mathrm{q}\cdot\mathrm{R}}\frac{(N-n)n}{N(N-1)},
\tag{11}
$$

does not decay as $|\mathrm{R}|$ increases. The detail derivation is given in the Appendix A.

## 3 Coalescing condensate states

In parallel, without loss of generality, we have

$$
|\varphi_{n}\rangle = \frac{1}{\Omega_{n}}\left(\sum_{\mathrm{r}}\hat{a}_{\mathrm{r}}^{\dagger}e^{i\mathrm{q}\cdot\mathrm{r}}\right)^{n}|0\rangle,
\tag{12}
$$

for the equation

$$
H^{\dagger}\,|\varphi_{n}\rangle = n\sum_{\alpha=1}^{3}V_{\alpha}\,|\varphi_{m}\rangle,
\tag{13}
$$

which establishes the biorthonormal set $\{|\varphi_{m}\rangle, |\psi_{n}\rangle\}$, satisfying

$$
\langle\varphi_{m}|\psi_{n}\rangle = \delta_{mn},
\tag{14}
$$

except for some special cases. We start the demonstrations from the simplest case with $n = 1$. Straightforward derivation shows that

$$
\langle\varphi_{1}|\psi_{1}\rangle = 0,
\tag{15}
$$

if $q_{\alpha} = q_{\alpha}^{\mathrm{c}} = \pi m_{\alpha}/N_{\alpha}$ ($m_{\alpha} \in [1, 2N_{\alpha}-1]$, $m_{\alpha} \neq N_{\alpha}$) for any one of $\alpha$, which indicates that the complete set of eigenstates is spoiled. According to non-Hermitian quantum mechanics, the Hamiltonian with parameter $q_{\alpha}^{\mathrm{c}}$ has an EP, and $|\psi_{1}\rangle$ is referred to as a coalescing state. We note that an EP can be induced by the parameter along a single direction (any one of $\alpha = 1$, 2, and 3). In this sense, the conditions for occurrence of EP are independent of three directions. Then one can investigate the EP problem from a 1D system, which makes things easily accessible. However, it is not a straightforward conclusion that $|\psi_{n}\rangle$ is a coalescing state simultaneously, since operator $\hat{a}_{\mathrm{r}}^{\dagger}$ obeys an unusual commutation relations in Eq. (4).

Considering a 1D system with a set of Hamiltonians in Eq. (1) with open boundary condition, i.e., $N_1 = N$, $N_2 = N_3 = 1$, and $q_1^c = \pi m_1/N = 2\pi m_1/(2N)$ ($m_1 \in [1, 2N-1]$, $m_1 \neq N$), each Hamiltonian $H(q_1^c)$ is tuned at EP. The matrix representation of $H(q_1^c)$ in the single-particle invariant subspace should have an EP [42, 43] due to the existence of $2 \times 2$ Jordan block. A natural question is what happens in the $n$-particle invariant subspace and whether $|\psi_n\rangle$ is also a coalescing state. To answer this question, we consider another set of Hamiltonians in Eq. (1) with periodic boundary condition, i.e., $N_1 = 2N$, $N_2 = N_3 = 1$, and $q_1 = 2\pi m_1/(2N)$ ($m_1 \in [1, 2N]$). Each Hamiltonian $H(q_1)$ is Hermitian and supports the Schrodinger equation

$$H(q_1)|\Psi_n\rangle = nV_1|\Psi_n\rangle, \tag{16}$$

with eigenstates

$$|\Psi_n\rangle = \frac{1}{\Omega_n}\left(\sum_{m=1}^{2N}\hat{a}_{m e_1}^\dagger e^{-imq_1}\right)^n|0\rangle. \tag{17}$$

Notably, we find that the coalescing state $|\psi_1\rangle$ of $H(q_1^c)$ is exactly the half part of the eigenstate $|\Psi_1\rangle$. It is a starting point, based on which we can show that

$$\langle\varphi_n|\psi_n\rangle = 0, \tag{18}$$

i.e., state $|\psi_n\rangle$ is also a coalescing state of the Hamiltonian $H(q_1^c)$. The detail derivation is given in the Appendix B.

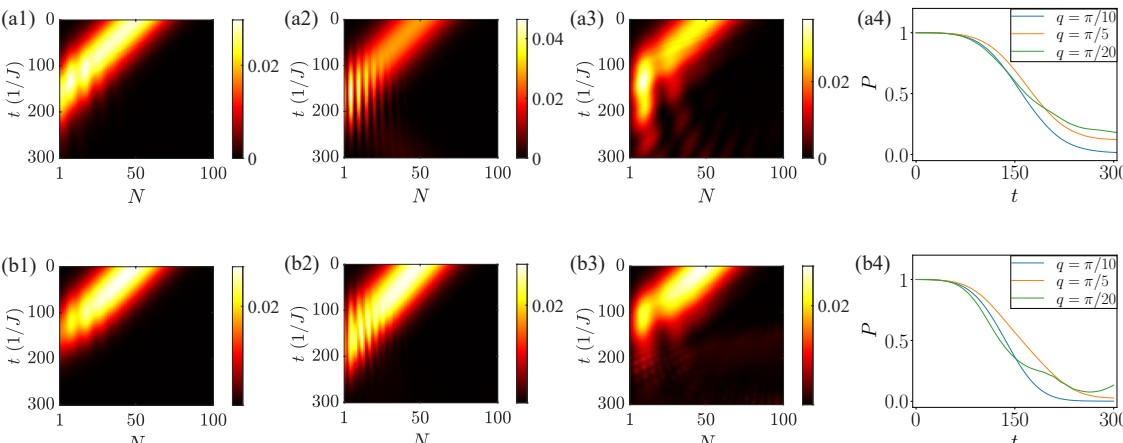

Figure 1: Plots of $p_j(t)$ and $P(t)$ defined in Eqs. (28) and (29) for the initial states defined in Eq. (27) with $n = 1$ in (a1)-(a3) and $n = 2$ in (b1)-(b3), respectively. The parameters of the Gaussian wavepacket are $\alpha = 0.05$, $N_0 = 50$ and $q = \pi/10$ in (a1, b1), $\pi/5$ in (a2, b2), $\pi/20$ in (a3, b3). The system parameter is $q_c = \pi/10$. The profiles of $p_j(t)$ in (a2, b2, a3, b3) exhibit evident interference fringes indicating reflections from the end of the chain. The plots of $P(t)$ indicate the perfect probability absorption for the resonant incident wavepackets.

## 4 Resonant scattering of non-Hermitian impurity

In this section, we focus on our study on 1D system for simplicity. The obtained result can be extended to 2D and 3D systems. We start with our investigation on the Hamiltonian in a

single-particle invariant subspace, in which the single-particle dynamics obeys a free boson model with the $\mathcal{PT}$ symmetric non-Hermitian Hamiltonian

$$H_{\text{FB}} = \sum_{j=1}^{N-1} \left( \hat{b}_j^\dagger \hat{b}_{j+1} + \text{H.c.} \right) + e^{-iq} \hat{b}_1^\dagger \hat{b}_1 + e^{iq} \hat{b}_N^\dagger \hat{b}_N, \tag{19}$$

on an $N$-site chain with complex on-site potential at two ends. Here $b_j^\dagger$ and $b_j$ are creation and annihilation operators for a boson on site $j$. We take a dimensionless constant for the sake of simplicity. According to the analysis in last two sections, states

$$\left( \sum_{j=1}^{N} e^{iqj} \hat{b}_j^\dagger \right)^n |0\rangle, \tag{20}$$

are $n$-boson eigenstates of $H_{\text{FB}}$. In addition, these states with different $n$ are coalescing states under the condition $q = q_{\text{c}} = \pi m / N$ ($m \in [1, 2N-1]$, $m \neq N$). Furthermore, this constraint for $q_{\text{c}}$ is satisfied automatically in large $N$ limit. It has been shown that this fact has intimate connection to the reflectionless scattering problem [43] for the semi-infinite system

$$H_{\text{FB}}^\infty = \sum_{j=1}^{\infty} \left( \hat{b}_j^\dagger \hat{b}_{j+1} + \text{H.c.} \right) + e^{-iq_{\text{c}}} \hat{b}_1^\dagger \hat{b}_1, \tag{21}$$

with a complex impurity at the end. The dynamic demonstration of this exact result is the near-perfect reflectionless of a Gaussian wavepacket with resonant momentum $q = q_{\text{c}}$. Specifically, we consider an initial $n$-boson state in the form

$$|\phi(0)\rangle = \left( \sum_j g_j \hat{b}_j^\dagger \right)^n |0\rangle, \tag{22}$$

where the single-boson wave function has the form

$$g_j = e^{-\frac{\alpha^2}{2}(j-N_0)^2} e^{iqj}. \tag{23}$$

The shape and center of the wavepacket are determined by parameters $\alpha$ and $N_0$. The near-perfect reflectionless indicates the time evolution of $|\phi(0)\rangle$ obeys

$$\lim_{t \to \infty} e^{-iH_{\text{FB}}^\infty t} |\phi(0)\rangle \approx 0. \tag{24}$$

It holds true for any $n$ with small $\alpha$, due to the following facts. (i) A wider single-boson wavepacket with $q = q_{\text{c}}$ can reflect the exact result for plane wave scattering from the end [43]. (ii) Multi-boson wavepacket shares the same dynamic behavior of a single-boson, since there is no interaction between bosons.

Now we turn to the hardcore boson Hubbard model, by ruling out the double occupation and adding the resonant NN interaction with the Hamiltonian

$$\begin{aligned} H_{\text{HB}}^\infty &= \sum_{j=1}^{\infty} \left( \hat{a}_j^\dagger \hat{a}_{j+1} + \text{H.c.} + \cos q_{\text{c}} \hat{n}_j \hat{n}_{j+1} \right) \\ &\quad + e^{-iq_{\text{c}}} \hat{a}_1^\dagger \hat{a}_1. \end{aligned} \tag{25}$$

The question is whether we still have the result

$$\lim_{t \to \infty} e^{-iH_{\text{HB}}^\infty t} \left( \sum_j g_j \hat{a}_j^\dagger \right)^n |0\rangle \approx 0, \tag{26}$$

Table 1: The structures of energy levels for 10-site open chain with different $q$ and filling particle number $n$. We list the numbers of coalescing states $n_{CS}$, of order $n_{OR}$ and the numbers of complex levels $n_{CM}$ in the form $(n_{CM}, n_{OR} \times n_{CS})$. It indicates that all the energy levels are real and all the systems contain a single 2 order coalescing state.

| q | n=2 | 3 | 4 | 5 |
|---|---|---|---|---|
| $\pi/10$ | $0, 2 \times 1$ | $0, 2 \times 1$ | $0, 2 \times 1$ | $0, 2 \times 1$ |
| $2\pi/10$ | $0, 2 \times 1$ | $0, 2 \times 1$ | $0, 2 \times 36$ | $0, 2 \times 43$ |
| $3\pi/10$ | $0, 2 \times 1$ | $0, 2 \times 1$ | $0, 2 \times 1$ | $0, 2 \times 1$ |
| $4\pi/10$ | $0, 2 \times 1$ | $0, 2 \times 1$ | $0, 2 \times 36$ | $0, 2 \times 43$ |

Table 2: The same as Table 1 but for $N_1 \times N_2 = 5 \times 3$ lattice with different $(q_1, q_2)$. We take open boundary condition in $q_1$-direction and periodic boundary condition in $q_2$-direction. It indicates that all the systems contain complex levels and multiple 2 order coalescing states.

| $(q_1, q_2)$ | n=2 | 3 | 4 |
|---|---|---|---|
| $(\pi/5, 2\pi/3)$ | $7, 2 \times 11$ | $135, 2 \times 5$ | $525, 2 \times 2$ |
| $(2\pi/5, 2\pi/3)$ | $16, 2 \times 11$ | $173, 2 \times 5$ | $586, 2 \times 2$ |
| $(3\pi/5, 2\pi/3)$ | $16, 2 \times 11$ | $171, 2 \times 5$ | $586, 2 \times 2$ |
| $(4\pi/5, 2\pi/3)$ | $13, 2 \times 11$ | $159, 2 \times 5$ | $570, 2 \times 2$ |

for the initial state

$$|\phi(0)\rangle = \left(\sum_j g_j \hat{a}_j^\dagger\right)^n |0\rangle. \tag{27}$$

To answer this question, numerical simulations are performed for the $n$-hardcore-boson initial wavepackets $|\phi(0)\rangle$ with $q$ around $q_c$. The profiles of the evolved states and the total probabilities are measured by

$$p_j(t) = \left|\frac{\hat{a}_j |\phi(t)\rangle}{|\phi(0)\rangle}\right|^2, \tag{28}$$

and

$$P(t) = \frac{1}{n}\sum_j p_j(t) = \left|\left|\frac{|\phi(t)\rangle}{|\phi(0)\rangle}\right|\right|^2, \tag{29}$$

respectively. We plot the two quantities in Fig. 1(a) and (b). The parameters of the initial wavepackets and the driven system are given in the captions. The profiles of eveloved states and the total probabilities indicate the perfect absorption for the resonant incident wavepackets with $q = q_c$ for both one- and two-boson cases, in accord with the previous theoretical analysis.

# 5 Dynamic generation of condensate states

Condensate state as macroscopic quantum state has significance both in theoretical and experimental physics. The stability of such states is crucial in practice. In general, it can be prepared as ground state by discreasing the temperature. Dynamical generation of nonequilibrium steady condensate states is another way, which has been received much attention recently. The advantage of an EP system is that a coalescing state is also a target state for a

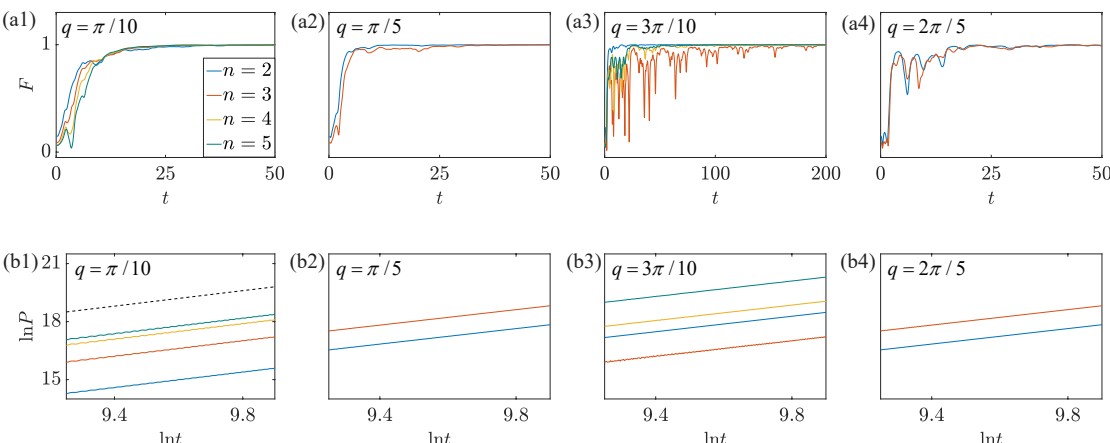

Figure 2: (a1)-(a4) present the time evolution of fidelity of $n$-particle one-dimensional system with $q = m\pi/N$ where m ranges from 1 to 4. (b1)-(b4) present the time evolution of probability of the same system. The slope of dashed line as reference is 2 and it is nearly parallel to other colored lines. Here we only plot the results with a single 2 order coalescing state. The initial state is $\prod_{j=1}^{n} \hat{a}_j^\dagger |0\rangle$ and the size of system $N = 10$, where the number n is indicated in the panel.

long-time evolution of various initial states. In the above, we have shown that the state $|\psi_n\rangle$ is also a coalescing state of the Hamiltonian $H(q_1^c)$. However, the order of EP is unknown, maybe depends on the particle numbers $n$, and the numbers of dimensions $n_d$ at which the resonant boundary is taken.

In contrast, the order of EP for free boson model can be exactly obtained. In a single-boson invariant subspace, the maximal order of EP is $n_d$, and then is $m$ for $n$-boson system, where

$$m = \frac{(n_d + n - 1)!}{n!(n_d - 1)!}. \tag{30}$$

According to the non-Hermitian quantum theory, there is a $m$-D Jordan block $M$ in the matrix representation of the Hamiltonian, which ensures that

$$M^m = 0. \tag{31}$$

The dynamics for any state in this subspace of Jordan block, referred to as auxiliary states, is governed by the time evolution operator

$$U(t) = e^{-iEt} e^{-iMt} = e^{-iEt} \sum_{l=0}^{m-1} \frac{1}{l!} (-iMt)^l, \tag{32}$$

where $E$ is a constant without any effect on the evolved state. It indicates that for an initial state $|\phi(0)\rangle$ involving the auxiliary states we have $|\phi(t)\rangle = U(t)|\phi(0)\rangle$

$$\lim_{t\to\infty} |\phi(t)\rangle \propto \left( \sum_r e^{-i\mathbf{q}\cdot\mathbf{r}} \hat{b}_r^\dagger \right)^n |0\rangle, \tag{33}$$

within large $t$ region and

$$||\phi(t)\rangle|^2 \propto t^{2(m-1)}, \tag{34}$$

which is also a dynamic demonstration for the order of the Jordan block.

However, such an analysis may be invalid due to the particle-particle interactions, i.e., replacing $\hat{b}_r^\dagger$ by $\hat{a}_r^\dagger$. In this situation, numerical simulations for finite size systems can shed some light on the dynamic generation of the condensate states of hardcore bosons.

We perform numerical simulations on finite systems with the following considerations. (i) The analysis above only predicts the results within large time domain. The efficiency of the scheme should be estimated from numerical simulations. (ii) The final state seems to be independent of the initial states, which can be simply unentangled $n$-boson initial states in the form $|\phi(0)\rangle = \prod_{\{l_1, l_2, \dots l_j, \dots l_n\}} \hat{a}_{l_j}^\dagger |0\rangle$. (iii) The order of EP can be observed by the dynamic process. The energy levels of many-particle non-Hermitian system are complicated, containing complex energy levels and multiple coalescing states, which may pose an obstacle in the calculation of time evolution. Our strategy has two steps. First, we find out the structure of energy level for sample systems. Second, select several coalescing states as target states. In Tables 1 and 2, the numbers of complex energy and coalescing levels are listed. We select several cases with single coalescing state to perform the computations.

The evolved states $|\phi(t)\rangle$ are computed by exact diagonalization for finite systems with several typical set of parameters. We focus on the Dirac probability $P(t)$ defined in Eq. (29) and the fidelity

$$F(t) = \frac{|\langle \psi_n | \phi(t)\rangle|^2}{\|\phi(t)\rangle\|^2}, \tag{35}$$

which is the measure of the distance between the final state and the condensate states $|\psi_n(q)\rangle$. We plot the fidelity $F(t)$ in Fig. 2 as function of $t$ for selected systems and particle numbers. We also plot the probability $\ln P(t)$ as function of $\ln t$ to demonstrate the EP dynamic behavior. Numerical results in Fig. 2 show that the unentangled initial states (see the caption) can indeed evolve to the corresponding target with high fidelity. On the other hand, it can be seen that the slopes of the lines in $\ln P(t)$ -$\ln t$ plane accord with the predicted values.

# 6  Summary

In summary, we have studied the Hermitian and non-Hermitian extended hardcore Bose-Hubbard model on one-, two-, and three-dimensional lattices. A set of exact eigenstates are constructed and have the following implications: (i) The strong on-site repulsion and nearest neighboring interaction cannot block the formation of BEC under the moderate particle density, when two interacting strengths are matched with each other. (ii) The solutions for Hermitian systems with periodic boundary condition are available for any given size, in which the momentum of the condensate is nothing but the reciprocal vector. Then the resonant non-Hermitian impurities can result in coalescing hardcore-boson condensate states. As an alternative stable state beyond the ground state, a coalescing state may be obtained via natural time evolution, although it is also the excited eigenstate of the system. In this sense, our finding not only reveals the possible condensation of interaction bosons, but also provides a method for condensate state engineering in an alternative way.

# Acknowledgements

**Funding information**    This work was supported by the National Natural Science Foundation of China (under Grant No. 12374461).

# A  Condensate eigenstates with ODLRO

In this appendix, we present the derivations on the eigenstates of the Hamiltonian in Eq. (1) and resonant conditions for many-body coalescing states. Consider the state

$$|\psi_n\rangle = \frac{1}{\Omega_n} \left( [e^{-iq\cdot r}\hat{a}_r^\dagger + e^{-iq\cdot(r+R)}\hat{a}_{r+R}^\dagger] + A \right)^n |0\rangle , \tag{A.1}$$

where $A$ is an operator which does not contain $\hat{a}_r^\dagger$ and $\hat{a}_{r+R}^\dagger$. We have

$$|\psi_n\rangle = \frac{1}{\Omega_n} \sum_{k=1}^n C_n^k A^{n-k} [e^{-iq\cdot r}\hat{a}_r^\dagger + e^{-iq\cdot(r+R)}\hat{a}_{r+R}^\dagger]^n |0\rangle . \tag{A.2}$$

In fact, we note that

$$[e^{-iq\cdot r}\hat{a}_r^\dagger + e^{-iq\cdot(r+R)}\hat{a}_{r+R}^\dagger]^2 |0\rangle = 2e^{-iq\cdot(2r+R)}\hat{a}_r^\dagger \hat{a}_{r+R}^\dagger |0\rangle , \tag{A.3}$$

but

$$[e^{-iq\cdot r}\hat{a}_r^\dagger + e^{-iq\cdot(r+R)}\hat{a}_{r+R}^\dagger]^k |0\rangle = 0, (k > 2). \tag{A.4}$$

Then

$$|\psi_n\rangle = \frac{1}{\Omega_n} \{ A^n + nA^{n-1}[e^{-iq\cdot r}\hat{a}_r^\dagger + e^{-iq\cdot(r+R)}\hat{a}_{r+R}^\dagger] $$
$$+ e^{-iq\cdot(2r+R)}n(n-1)A^{n-2}\hat{a}_r^\dagger \hat{a}_{r+R}^\dagger \} |0\rangle . \tag{A.5}$$

One can take $R = e_\alpha$, and then we have

$$h_r^\alpha |\psi_n\rangle = 0, \tag{A.6}$$

which ensures that

$$]H |\psi_n\rangle = n \sum_{\alpha=1}^3 V_\alpha |\psi_n\rangle . \tag{A.7}$$

Furthermore, we have

$$\hat{a}_r^\dagger \hat{a}_{r+R} |\psi_n\rangle = \frac{1}{\Omega_n} \{ nA^{n-1}[e^{-iq\cdot r}\hat{a}_r^\dagger \hat{a}_{r+R}\hat{a}_r^\dagger $$
$$+ e^{-iq\cdot(r+R)}\hat{a}_r^\dagger \hat{a}_{r+R}\hat{a}_{r+R}^\dagger] \} |0\rangle $$
$$= \frac{1}{\Omega_n} nA^{n-1}e^{-iq\cdot(r+R)}\hat{a}_r^\dagger |0\rangle , \tag{A.8}$$

which results in the correlation function

$$\langle\psi_n| \hat{a}_r^\dagger \hat{a}_{r+R} |\psi_n\rangle = \left(\frac{n}{\Omega_n}\right)^2 e^{-iq\cdot R} \left| A^{n-1} |0\rangle \right|^2 $$
$$= e^{-iq\cdot R} \frac{(N-n)n}{N(N-1)}. \tag{A.9}$$

We find that

$$\lim_{|R|\to\infty} \left| \langle\psi_n| \hat{a}_r^\dagger \hat{a}_{r+R} |\psi_n\rangle \right| = \frac{n(N_1 N_2 N_3 - n)}{N_1 N_2 N_3 (N_1 N_2 N_3 - 1)}, \tag{A.10}$$

which is finite number, indicating off-diagonal long-range order (ODLRO) according to [41].

# B   Coalescing condensate states

We start with the case with $n = 1$. The biorthogonal norm for state $|\psi_1\rangle$ is

$$
\begin{aligned}
\langle \varphi_1 | \psi_1 \rangle &= \frac{1}{\Omega_n^2} \sum_r e^{-i2\mathbf{q}\cdot\mathbf{r}} \\
&= \frac{1}{\Omega_n^2} \prod_{\alpha=1,2,3} \sum_{m_\alpha} e^{-i2q_\alpha m_\alpha}.
\end{aligned}
\tag{B.1}
$$

We note that if

$$
\sum_{m_\alpha} e^{-i2q_\alpha m_\alpha} = 0,
\tag{B.2}
$$

for one of $\alpha$, we have

$$
\langle \varphi_1 | \psi_1 \rangle = 0,
\tag{B.3}
$$

i.e., an EP can be induced by the parameter along a single direction.

In the following, we will show that it is also true for the case with $n > 1$. We only consider 1D systems for the sake of simplicity. We focus on two Hamiltonians. The first one is

$$
\begin{aligned}
H_{\text{HB}}^1 = &\sum_{j=1}^{N-1} \left( \hat{a}_j^\dagger \hat{a}_{j+1} + \text{H.c.} + \cos q \, \hat{n}_j \hat{n}_{j+1} \right) \\
&+ e^{-iq} \hat{a}_1^\dagger \hat{a}_1 + e^{iq} \hat{a}_N^\dagger \hat{a}_N,
\end{aligned}
\tag{B.4}
$$

which is non-Hermitian and reduced from Eq. (1) for an $N$-site chain. The second one is

$$
H_{\text{HB}}^2 = \sum_{j=1}^{2N} \left( \hat{a}_j^\dagger \hat{a}_{j+1} + \text{H.c.} + \cos q \, \hat{n}_j \hat{n}_{j+1} \right),
\tag{B.5}
$$

which is Hermitian and reduced from Eq. (1) for a $2N$-site ring. Defining a set of collective operators

$$
A^+ = \frac{1}{\sqrt{N}} \sum_{j=1}^{N} \hat{a}_j^\dagger e^{-iqj}, \quad B^+ = \frac{1}{\sqrt{N}} \sum_{j=N+1}^{2N} \hat{a}_j^\dagger e^{-iqj},
\tag{B.6}
$$

and

$$
A^- = \frac{1}{\sqrt{N}} \sum_{j=1}^{N} \hat{a}_j e^{-iqj}, \quad B^- = \frac{1}{\sqrt{N}} \sum_{j=N+1}^{2N} \hat{a}_j e^{-iqj},
\tag{B.7}
$$

with $q = 2\pi m/(2N)$ ($m \in [1, 2N-1]$, $m \neq N$), a subset of the eigenstates of $H_{\text{HB}}^1$ can be expressed as

$$
|\psi_n\rangle = \frac{1}{\Lambda_n} \left( A^+ \right)^n |0\rangle,
\tag{B.8}
$$

while states

$$
|\Psi_n\rangle = \frac{1}{\Lambda_n'} \left( A^+ + B^+ \right)^n |0\rangle,
\tag{B.9}
$$

and

$$
|\Psi_n^*\rangle = \frac{1}{\Omega_n'} \left[ \left( A^+ + B^+ \right)^* \right]^n |0\rangle,
\tag{B.10}
$$

are a subset of the eigenstates of $H_{\text{HB}}^2$, here

$$
\Lambda_n = \frac{N^{n/2}}{(n!)\sqrt{C_N^n}}, \quad \Lambda_n' = \frac{N^{n/2}}{(n!)\sqrt{C_{2N}^n}}.
\tag{B.11}
$$

We note that state $|\psi_n\rangle$ is a part of state $|\Psi_n\rangle$, which is crucial for the following proof. The orthogonality of two states $|\Psi_n\rangle$ and $\left|\Psi_n^*\right\rangle$ leads to

$$
\begin{aligned}
&\langle\Psi_n^*|\Psi_n\rangle \\
&= \sum_{m=0}^{n} p_m \langle 0|\left(A^-\right)^{n-m}\left(B^-\right)^m\left(A^+\right)^{n-m}\left(B^+\right)^m|0\rangle \\
&= 0.
\end{aligned} \tag{B.12}
$$

Taking $n = 1$, we have

$$
\langle 0|\left(A^-A^+ + B^-B^+\right)|0\rangle = 0, \tag{B.13}
$$

which resuts in

$$
\langle 0|A^-A^+|0\rangle = \langle 0|B^-B^+|0\rangle = 0, \tag{B.14}
$$

due to the translational symmetry of state $|\Psi_n\rangle$. Based on this conclusion, taking $n = 2$, we have $\langle 0|\left(B^-B^+\right)^2|0\rangle = \langle 0|\left(A^-A^+\right)^2|0\rangle = 0$. Furthermore, it turns out that

$$
\langle 0|\left(A^-A^+\right)^m|0\rangle = 0, \tag{B.15}
$$

for $m \in [0, n]$, which results in

$$
\langle\varphi_n|\psi_n\rangle = 0. \tag{B.16}
$$

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
