# Peer review of "Coalescing hardcore-boson condensate states with nonzero momentum"

_SciPost Physics Core_

## Round 1 · Referee Report · Anonymous (Referee 1) · 2024-10-7

Strengths

1- The idea of ​​looking for ways to obtain condensates with non-zero momentum. 2- Analytical and numerical calculations

Weaknesses

1- Despite the two appendices, the presentation of the analytical calculations is short and confusing, difficult to follow. 2- The discussion of the figures was short and vague.

Report

In the manuscript untitled "Coalescing hardcore-boson condensate states with nonzero momentum" by C. H. Zhang and Z. Song, the authors made an analitical and numerical study of the Hermitian and non-Hermitian extended hardcore Bose-Hubbard model on one-, two- and three-dimensional lattices. The authors claimed that there are condensate modes when the system parameters meet the matching conditions, and show that the correlations functions possesses off-diagonal long-rangfe order (ODLRO). Also they discussed the impact of non-Hermitian impurities on the dynamic formation of condensate states, which become coalescing states when the non-Hermitian PT-symmetric boundary induces the exceptional points.
The paper sounds interesting, however it is very difficult to read. The presentation is short and confusing; I do not understand why the authors limit themselves to discussing the physics involved in the paper. For example, the discussion of the figures is extremely short and not very explanatory. As for the analytical results, these seem correct; however, additional tests are required.
Based on the above reasons, I believe that this paper required major corrections to deserve publication in Scipost Physics Core.

Requested changes

1- Expand the discussion of the analytical results, figures, and physics involved in the paper.

Recommendation

Ask for major revision

---

## Editorial Decision

resubmitted